# Contact-dependent traits in *Pseudomonas syringae* B728a

**Monica N. Hernandez**[¤]*, **Steven E. Lindow**

Department of Plant and Microbial Biology, University of California, Berkeley, California, United States of America

¤ Current address: Department of Botany and Plant Pathology, Southern Oregon Research and Extension Center affiliated with Oregon State University, Central Point, Oregon, United States of America
* monica.hernandez@oregonstate.edu

**Data Availability Statement:** All processed and raw RNA Seq data files are available from the Gene Expression Omnibus (GEO) database (accession number(s) GSE157877).

**Funding:** This work used the Vincent J. Coates Genomics Sequencing Laboratory at UC Berkeley,

## Abstract

Production of the biosurfactant syringafactin by the plant pathogen *Pseudomonas syringae* B728a is a surface contact-dependent trait. Expression of *syfA*, as measured using a *gfp* reporter gene fusion was low in planktonic cells in liquid cultures but over 4-fold higher in cells immobilized on surfaces as varied as glass, plastic, paper, parafilm, agar, membrane filters, and leaves. Induction of *syfA* as measured by GFP fluorescence was rapid, occurring within two hours after immobilization of cells on surfaces. Comparison of the global transcriptome by RNA sequencing of planktonic cells in a nutrient medium with that of cells immobilized for 2 hours on filters placed on this solidified medium revealed that, in addition to *syfA*, 3156 other genes were differentially expressed. Genes repressed in immobilized cells included those involved in quaternary ammonium compound (QAC) metabolism and transport, compatible solute production, carbohydrate metabolism and transport, organic acid metabolism and transport, phytotoxin synthesis and transport, amino acid metabolism and transport, and secondary metabolism. Genes induced in immobilized cells included *syfA* plus those involved in translation, siderophore synthesis and transport, nucleotide metabolism and transport, flagellar synthesis and motility, lipopolysaccharide (LPS) synthesis and transport, energy generation, transcription, chemosensing and chemotaxis, replication and DNA repair, iron-sulfur proteins, peptidoglycan/cell wall polymers, terpenoid backbone synthesis, iron metabolism and transport, and cell division. That many genes are rapidly differentially expressed upon transfer of cells from a planktonic to an immobilized state suggests that cells experience the two environments differently. It seems possible that surface contact initiates anticipatory changes in *P. syringae* gene expression, which enables rapid and appropriate physiological responses to the different environmental conditions such as might occur in a biofilm. Such responses could help cells survive transitions from aquatic habitats fostering planktonic traits to attachment on surfaces, conditions that alternatively occur on leaves.

supported by NIH S10 OD018174 Instrumentation Grant. This work was also supported by the National Science Foundation Louis Stokes Alliances for Minority Participation Bridge to the Doctorate Fellowship (https://www.nsf.gov/awardsearch/showAward?AWD_ID=1249249), the Chancellor's Fellowship for Graduate Study (https://grad.berkeley.edu/admissions/apply/fellowships-entering/), and the William Carroll Smith Fellowship. These fellowships were awarded to MH. The funders had no role in study design, data collection and analysis, decision to publish, or preparation of the manuscript.

**Competing interests:** The authors have declared that no competing interests exist.

## Introduction

*Pseudomonas syringae* has adapted to live in a variety of different environments. While most studies of this taxon have focused on its life as a plant colonist in which it grows on the surfaces of leaves and subsequently in the apoplast where it can cause disease, many strains apparently inhabit other habitats. For instance, Morris *et al*. [1] described *P. syringae* in snow, rain, and in lakes and rivers, suggesting that this bacterium is disseminated through the water cycle. All bacteria are intrinsically aquatic, although many have adapted to survive in periodically dry environments, often forming biofilms on surfaces in contact with water. The very different chemical and physical properties of aquatic environments versus surfaces, which in the case of *P. syringae* are frequently dry, presumably select for coordinated gene expression that optimizes fitness across these different conditions. Leaf surfaces are relatively harsh habitats that exhibit strong spatially and temporally varying conditions. Water and nutrients are unevenly dispersed across leaf surfaces [2, 3] and leaves experience rapid temporal fluctuations in temperature, humidity, and liquid water availability [3]. Leaf surfaces also experience high ultraviolet radiation flux [3]. In a bacterial population inhabiting such a setting, some cells might exhibit a planktonic existence within water drops containing nutrients, while other cells must contend with relatively dry surfaces.

Previous studies have revealed the role of syringafactin, a hygroscopic biosurfactant whose production by *P. syringae* is encoded by *syfA*, as an adaptation to the frequent absence of liquid water on leaf surfaces [4, 5]. By binding water vapor trapped within the humid boundary layer and/or by binding liquid water that is transiently present on leaves, syringafactin can alleviate water stress by expanding the time frame during which liquid water is available for *P. syringae* [6]. Furthermore, syringafactin does not seem to disperse in water, perhaps due to its strong hydrophobic characteristics, but instead remains immobilized onto leaf surfaces near the cells that produced it [5]. Syringafactin is thus unlikely to benefit cells immersed in water, such as those occurring in the planktonic state in aquatic environments involved in the water cycle [1]. Instead, syringafactin production might maximally benefit cells in non-aquatic settings where it would improve *P. syringae* survival on surfaces subject to frequent drying. Interestingly, Burch *et al*. [7] found that cells grown on agar surfaces produced much more syringafactin and expressed *syfA* at a much higher level than those in broth cultures. This observation suggests two competing hypotheses. First, *syfA* differential gene expression may be instigated by the differences in chemical or physical conditions found on an agar surface compared to that in a similar liquid culture. Alternatively, immobilization of cells on a surface may be used as a cue to immediately indicate the current or anticipated presence of environmental conditions common to leaf surfaces where the production of syringafactin would be maximally beneficial to the cell. Since Burch *et al*. [7] examined *syfA* expression only after growth of bacteria in different conditions for extended time periods (>24 hours), they could not unambiguously distinguish between these two models of gene regulation.

To distinguish between these two hypotheses, we determined how quickly *syfA* expression increased after planktonic cells of *P. syringae* contacted a surface and whether the nature of the surface it interacted with determined the extent of any response. It seemed likely that cells of *P. syringae* might benefit from anticipatory patterns of gene expression, where certain cues might be used to indicate large and rapid changes in environmental conditions. Thus, rapid or prior expression of certain traits would be highly beneficial in adapting to large changes in habitat characteristics. In this case, we predicted that changes in gene expression associated with transition from a planktonic to a surface-associated habitat such as a leaf would be linked to perception of a surface that might be imminently dry. While there has been interest in such surface sensing in bacteria [8–12] little is known of the mechanism by which cells sense

surfaces and there are few studies that have examined rapid changes in gene expression that could be associated with perception of a surface [13]. We therefore explored the rapid changes in the global transcriptome of B728a soon after planktonic cells became sessile on a surface. The large number of genes that we find to undergo changes in gene expression upon the transition from planktonic to sessile life suggest that surface sensing could be a major cue controlling adjustment of *P. syringae* to the varying habitats it colonizes on leaf surfaces and elsewhere.

## Materials and methods

### Bacterial strains and growth conditions

*Pseudomonas syringae* B728a strains were grown on King's medium B (KB) plates containing 1.5% technical agar or in KB broth lacking agar [14]. Antibiotics were used at the following concentrations (μg/ml): spectinomycin (100), kanamycin (50), and tetracycline (15) as appropriate.

### Quantification of GFP fluorescence in individual bacterial cells

*P. syringae* B728a cells harboring a plasmid in which the *syfA* promoter was fused to a promoterless *gfp* reporter gene [7], were grown in KB broth until they reached a density of $10^8$ cells/ml as determined by Optical Density ($\lambda = 600$ nm). 10 μl of cells were then placed onto various surfaces. To subsequently remove cells from these surfaces, moistened cotton swabs were used to scrub off the cells. The swabs were then placed into microcentrifuge tubes containing 100 μl of water and vortexed to re-suspend the cells. 5 μl of cell suspension from each treatment was applied to glass slides for fluorescence microscopy to quantify the GFP fluorescence of individual cells. Cells were visualized at 100X magnification with a M2 AxioImager. A GFP filter set was used to view cells, and images were captured in black and white format using a 12-bit Retiga camera. The software iVision was used to identify all bacterial cells in an image and to quantify the average pixel intensity of each object identified as in other studies [6]. Aggregates of bacterial cells and extraneous particles were identified by visual examination and marked for exclusion before image processing.

### RNA isolation

Bacterial cells were grown with shaking in KB broth at 28˚C until cultures reached a cell density of $5x10^8$ cells/ml. Three replicate cultures were used. 300 μl of each culture was then applied to a 0.4 μm Isopore® membrane filter and excess liquid was removed by exposing the filter to a mild vacuum for a very short period of time ($< 2$ seconds) only until the liquid had been removed. Filters were then immediately placed onto KB plates to incubate at 28˚C for two hours while broth cultures were returned to a shaker and incubated at 28˚C for two more hours. Cells in broth cultures were harvested by pipetting 1 ml of suspension into a 15 mL conical tube containing 125 μl ice-cold EtOH/Phenol stop solution (5% water-saturated phenol (pH<7.0) in ethanol). Filters were immersed in the EtOH/Phenol stop solution and cells were harvested by sonication for 30 seconds followed by vortexing for 20 seconds to ensure complete cell detachment from the filters. Cells were then collected by centrifugation at 12,000 rpm (13,800 x g) for five minutes at 4˚C. Supernatant was decanted and the cells were frozen in liquid nitrogen and stored at -80˚C until RNA isolation. RNA isolation was performed using a Direct-zol™ RNA Kit (Zymo Research) and followed the manufacturer's instructions. RNA was stored frozen at -80˚C.

## mRNA sequencing

1 μl of each sample was diluted into 4 μl of RNase free water and submitted to the Vincent J. Coates Genomics Sequencing Laboratory at UC Berkeley where Ribo-Zero was used for rRNA removal. RNA abundance and purity were determined using a 2100 Bioanalyzer (Agilent Technologies) and quantified using Qubit (Invitrogen). After reverse transcription, shearing of cDNA, size fractionation, and Illumina library production, the Vincent J. Coates Genomics Sequencing Laboratory samples were sequenced using an Illumina HiSeq4000 platform with 50 base pair, single-end reads. Three biological replicates were sequenced per treatment. Reads were uploaded to Galaxy [15] and cleaned using Trimmomatic [16]. Reads were aligned to the *Pseudomonas syringae* B728a genome [17] using Salmon Transcript Quantification [18] in Galaxy. The program edgeR in R was then used to assess the statistical significance of differential gene expression [19]. Gene expression levels were normalized using a weighted trimmed mean of M values (TMM; where M is the log expression ratio per gene between treatments) [20]. Empirical Bayes estimation and tests based on the negative binomial distribution were then used to determine significance [20]. The Fisher's exact test was used to make comparisons between groups. This test gave values for the log fold change and the log counts per million as well as the p-value and the false discovery rate for each gene. To account for the false discovery rate, genes that had a false discovery rate of <0.05 were selected for further analyses. A gene was considered significantly differentially regulated if the p-value for the difference in relative expression between the filter treatment and the liquid treatment was less than 0.001.

## Statistical analysis

The hypergeometric distribution was performed in R [21] to test for significance of functional category enrichment [22]. All p-values were adjusted using the Bonferroni correction [23] and the Benjamini-Hochberg correction [24] in R. Mean GFP measures were compared using the Tukey HSD test in R.

## Results

### *SyfA* induction occurs rapidly upon surface contact

If the *syfA* gene is regulated directly as a response to contact with a surface, we would expect gene induction to occur shortly after surface contact. To test how rapidly *syfA* was induced after planktonic cells were transferred to various surfaces, we monitored the expression of *syfA* in *P. syringae* B728a harboring a plasmid containing the *syfA* promoter fused to a promoterless *gfp* reporter gene by assessing the GFP fluorescence of individual cells by epifluorescence microscopy. Planktonic cells from broth cultures were applied to three surfaces: (1) agar-solidified King's medium B (KB) media; (2) a 0.4 μm polycarbonate Isopore® filter that was placed on KB agar; and (3) a 0.4 μm Isopore® filter, that was moistened by KB broth, and which was then placed on a plastic surface. Since nutrient solution diffused through the filters, wetting the filters in these settings, we presume that the cells experienced similar levels of nutrients while immobilized on the filters as those remaining in broth cultures but experienced different physical conditions. Planktonic cells that remained in broth cultures served as a control. We compared gene expression of cells on filters placed on KB agar and KB broth to test whether chemical components of the agar, rather than simply the physical change it elicited, contributed to induction of *syfA* expression as seen by Burch *et al.* [7]. By two hours after transfer to each of the solid surfaces, cells exhibited significantly greater GFP fluorescence than those remaining in the broth cultures (Fig 1). Similar levels of *syfA* induction occurred on all surfaces (Fig 1). Thus, *syfA* induction clearly occurs rapidly upon encountering a surface and elevated

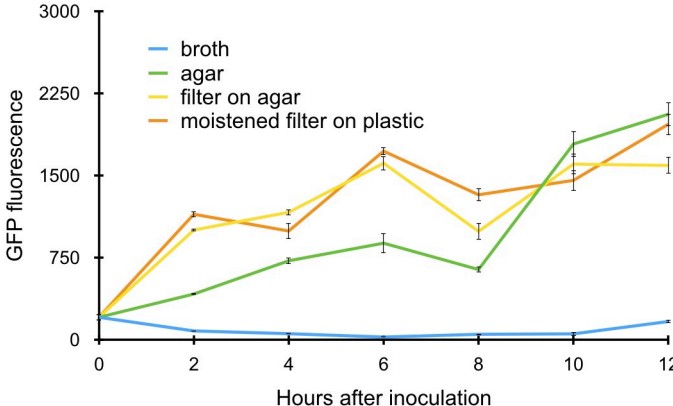

**Fig 1. Induction of *syfA* expression in cells of *Pseudomonas syringae* B728a as a function of time.** Gene expression is estimated as the mean GFP fluorescence of individual cells harboring a plasmid in which the *syfA* promoter was fused to a *gfp* reporter gene when harvested from these conditions at the various times shown on the abscissa. 7,038 cells were evaluated in total for control cells that remained in a broth culture (blue); 3,068 cells were evaluated after application to an agar surface (green); 10,184 cells were evaluated after application to filters placed on an agar surface (yellow); and 3,791 cells were evaluated after application to a filter moistened with KB broth that was then placed on plastic (orange). The vertical bars represent the standard error of the mean GFP fluorescence for a given cell.

expression on agar appears to be due to the exposure of cells to a surface rather than any chemical components of the agar.

## *SyfA* is induced in cells on a variety of solid surfaces

Given that *syfA* induction quickly followed transfer of planktonic cells to various surfaces with abundant nutrient resources, we explored whether any particular physical parameters such as hydrophobicity or roughness were associated with the induction process. Since *P. syringae* B728a was isolated from leaf surfaces and is highly fit as an epiphyte [25, 26], we tested the hypothesis that leaf surfaces would confer the most rapid or largest contact-dependent induction of this gene. By exposing cells to various types of solid surfaces we further explored whether *syfA* expression was regulated by contact with a surface per se as opposed to a chemical cue. Planktonic cells from KB broth cultures were applied to polycarbonate, paper, parafilm, excised leaves, water agar, and Isopore®, Durapore®, and Teflon® filters. Cells all exhibited at least 2-fold greater GFP fluorescence than those of the planktonic cells when assessed six hours after application to these surfaces (Fig 2). Little difference in GFP fluorescence was observed among cells placed on the various solid surfaces, suggesting that neither surface type-dependent chemical cues nor the physical properties of the surface on which cells were immobilized strongly influenced the induction of *syfA*.

## RNA sequencing of *P. syringae* B728a reveals many genes differentially regulated in cells transferred to a filter surface versus in liquid culture

Given that cells of *P. syringae* exhibited large increases in *syfA* expression within two hours of transfer to a variety of solid surfaces, we performed RNA sequencing to determine the extent to which other genes exhibit contact-dependent gene expression. The transcriptome of cells grown in KB broth was compared to those placed on filters atop KB agar for 2 hours. RNA sequencing returned between 20 and 50 million reads per sample (S1 Table). As expected, a wide range of reads were recovered for a given gene. A heatmap revealed that while patterns of

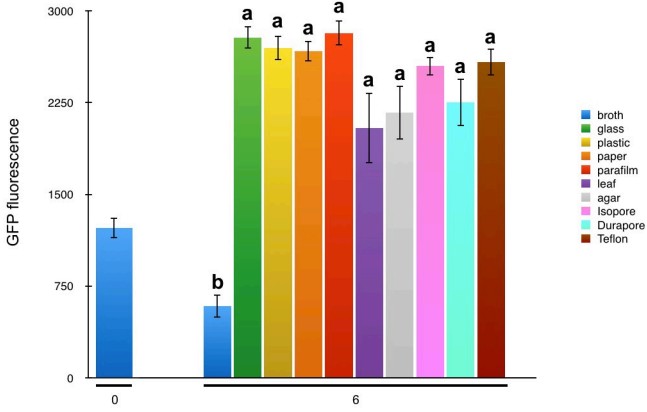

**Fig 2. The expression of *syfA* increases rapidly after immobilization of cells of *Pseudomonas syringae* B728a.** Gene expression is estimated as the mean GFP fluorescence of individual cells harboring a plasmid in which the *syfA* promoter was fused to a *gfp* reporter gene when harvested 6 hours after application to glass (green), polycarbonate plastic (yellow), paper (orange), parafilm (red), bean leaves (purple), agar (grey), Isopore filters (pink), Durapore filters (aqua), Teflon filters (brown), or recovered from broth cultures at the time of application to surfaces or after 6 hours (blue). The vertical bars represent the standard error of the mean GFP fluorescence for a given cell. Bars sharing a letter are not significantly different at p<0.05 by Tukey HSD test.

gene regulation were similar among replicates in which cells were either planktonic or immobilized on a filter, gene expression differed strongly between cells in these two conditions (Fig 3). Much of the variation in relative gene expression levels of the average gene was thus associated with the environment of the cells before RNA was harvested rather than other factors.

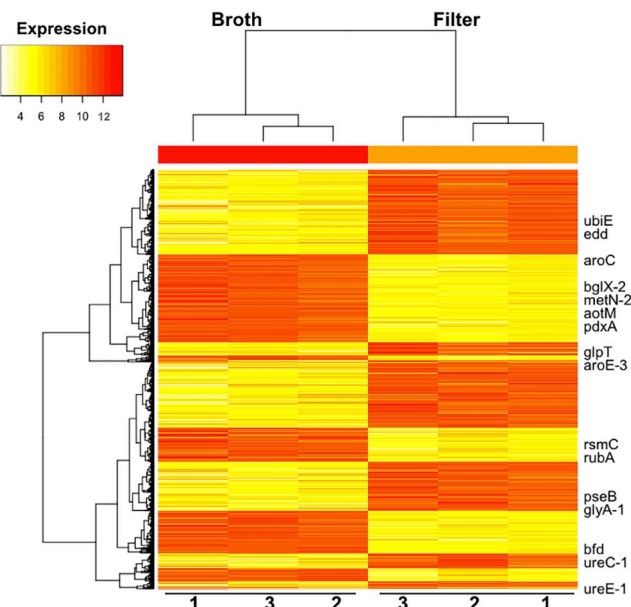

**Fig 3. Heat map illustrating different patterns of gene expression in cells of *Pseudomonas syringae* B728a.** Heat map illustrating different patterns of gene expression in cells of *Pseudomonas syringae* B728a 2 hours after application to a filter surface compared to that in broth culture. Red indicates high expression and yellow indicates low expression.

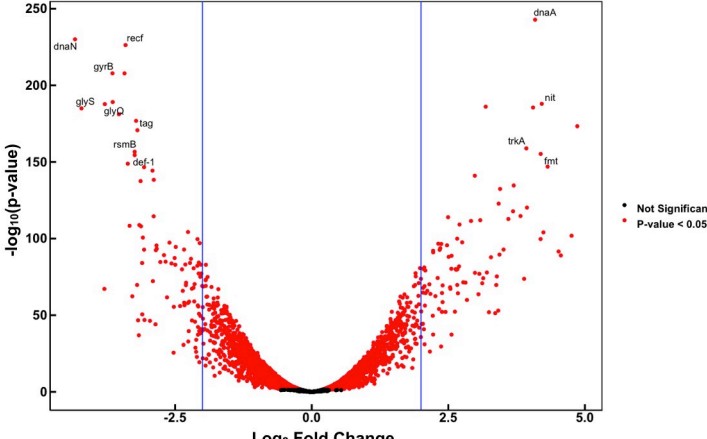

**Fig 4. Volcano plot illustrating the differential expression of genes in *Pseudomonas syringae* B728a.** Volcano plot illustrating the differential expression of genes in *Pseudomonas syringae* B728a 2 hours after application to a filter surface. Genes that are significantly differentially expressed compared to that in broth culture are shown in red, and those that do not differ are shown in black.

Expression of 1,390 genes was up-regulated in cells on the filter surface compared to those in broth media, while expression of 1,766 genes was down-regulated on filter surfaces compared to that in broth (Fig 4). Of the 1,390 genes that were induced on filter surfaces, 881 were induced more than 2-fold while 509 exhibited lesser induction. Of the 1,766 genes that were repressed on filter surfaces, 1,138 were repressed more than 2-fold while 628 were repressed less than 2-fold. In total, 3,156 genes (60.46% of the *P. syringae* B728a genome) were found to be differentially expressed in cells on the filter surface compared to those in broth culture (Fig 4). There was no apparent relationship between the absolute level of expression of a gene and the likelihood that it would exhibit differential expression in these two settings (S1 Fig).

## Functional category analysis

We grouped the differentially regulated genes into functional categories to better establish those processes that were most differentially expressed in planktonic and immobilized cells (Tables 1 and 2). Twice as many functional gene categories were enriched in genes that were significantly up-regulated in immobilized cells than those that were down-regulated. The proportion of genes in various gene categories for which there is statistical support for differential up-regulation (Fig 5A) or down-regulation (Fig 5B) of gene expression upon immobilization on membranes differed substantially. The following gene functional categories were examined in further detail:

**Translation.** Many genes encoding the 30S and 50S ribosomal protein subunits were induced on the filter surface, as were genes encoding the elongation factor proteins Ts, P, Tu, and G. Many genes encoding t-RNA synthetases were also induced. It thus appeared that translation as a whole may have accelerated upon transition of cells from a planktonic to an immobilized state.

**Siderophore synthesis and iron metabolism.** Many genes involved in siderophore synthesis and transport were induced on the filter surface. Many genes such as *pvdS*, *pvdG*, *pvdL*, *pvdI*, *pvdJ*, *pvdK*, *pvdD*, *pvdE*, *pbdO*, *pvdN*, *pvdT*, and *pvdR* were involved in regulation of pyoverdine production and its transport. Many genes involved in achromobactin regulation, synthesis, and transport including *acsG*, *acsD*, *acsE*, *yhcA*, *acsC*, *acsB*, *acsA*, *carA-2*, *cbrB-2*, and

**Table 1. Functional gene categories of *Pseudomonas syringae* B728a preferentially up-regulated in immobilized cells as compared to planktonic cells 2 hours after application to a filter surface.**

| Gene Category | Bonferroni adjusted p-value | Benjamini-Hochberg adjusted p-value |
|---|---|---|
| Translation | 1.11E-18 | 1.11E-18 |
| Siderophore synthesis and transport | 4.75E-09 | 2.38E-09 |
| Nucleotide metabolism and transport | 1.12E-06 | 3.73E-07 |
| Flagellar synthesis and motility | 2.44E-03 | 6.11E-04 |
| Lipopolysaccharide synthesis and transport | 9.16E-03 | 1.83E-03 |
| Energy generation | 0.02 | 2.63E-03 |
| Transcription | 0.04 | 0.04 |
| Chemosensing and chemotaxis | 0.06 | 7.15E-03 |
| Replication and DNA repair | 0.12 | 0.01 |
| Iron-sulfur proteins | 0.19 | 0.02 |
| Peptidoglycan/cell wall polymers | 0.19 | 0.02 |
| Terpenoid backbone synthesis | 0.24 | 0.02 |
| Iron metabolism and transport | 0.25 | 0.02 |
| Cell Division | 0.59 | 0.04 |

The significance of functional category enrichment was assessed using the hypergeometric distribution [22].

*cbrC-2* were also induced. Many genes involved in iron metabolism and transport were also induced on the filter surface. This included the genes *fecE*, *fecD*, *fecC*, *fecB*, *fecA*, *fecR*, and the RNA polymerase ECF sigma factor *fecI*. These results suggest that iron became less available on the filter surfaces than in broth, perhaps due to diffusional limitations associated with the lack of mixing of cells as would occur in a liquid medium. Alternatively, iron might commonly be less available on the natural surfaces on which *P. syringae* typically inhabits, such as leaf surfaces, and the filter mimicked physical cues that the bacteria might use to anticipate transition into such low iron environments.

**Nucleotide metabolism and transport.** Numerous genes involved in nucleotide metabolism and transport were induced in immobilized cells. This included many genes involved in purine and pyrimidine metabolism such as *purA*, *purT*, *purC*, *purF*, *purB*, *purM*, *purN*, *purU-3*, *purH*, *purD*, *purK*, *purE*, and *pyrB*, *pyrR*, *pyrH*, *pyrG*, *pyrF*, *pyrD*, *pyrC-2* respectively. As with the apparent increase in translational activity seen upon transition of planktonic cells to

**Table 2. Functional gene categories of *Pseudomonas syringae* B728a preferentially down-regulated in immobilized cells as compared to planktonic cells 2 hours after application to a filter surface.**

| Gene Category | Bonferroni adjusted p-value | Benjamini-Hochberg adjusted p-value |
|---|---|---|
| Quaternary ammonium compound metabolism and transport | 2.19E-08 | 2.19E-08 |
| Compatible solute synthesis | 2.33E-06 | 1.17E-06 |
| Carbohydrate metabolism and transport | 1.72E-05 | 5.74E-06 |
| Organic acid metabolism and transport | 8.84E-05 | 2.21E-05 |
| Phytotoxin synthesis and transport | 1.16E-04 | 2.32E-05 |
| Amino acid metabolism and transport | 0.01 | 1.77E-03 |
| Secondary metabolism | 0.01 | 1.01E-03 |

The significance of functional category enrichment was assessed using the hypergeometric distribution [22].

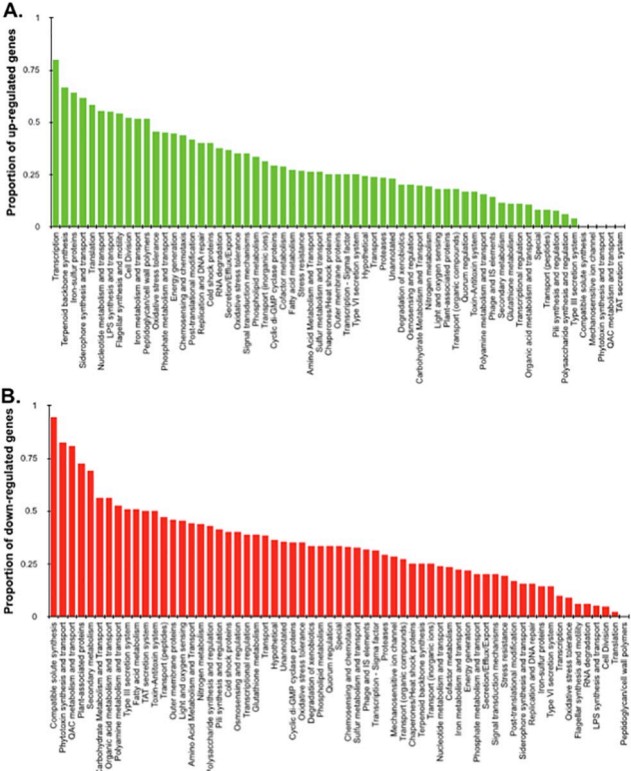

**Fig 5. Proportion of up-regulated and down-regulated genes of *Pseudomonas syringae* B728a.** Proportion of up-regulated and down-regulated genes of *Pseudomonas syringae* B728a in those functional gene categories that exhibited differential expression in immobilized cells as compared to planktonic cells 2 hours after application to a filter surface. (A) Proportion of up-regulated genes in each functional gene category and (B) proportion of down-regulated genes in each functional gene category.

those on surfaces, increased transcription might be expected to also be linked to such increases, requiring higher rates of nucleotide synthesis. Almost all genes with significant differential expression involved in cell division were also induced on the filter surface. This included the cell division proteins FtsK, FtsQ, and FtsL as well as the rod-shape determining proteins MreD and MreC. MrdB, a cell cycle protein, was also induced.

**Flagellar synthesis and motility.** Many genes encoding flagellar biosynthesis proteins, flagellar basal body proteins, and flagellar hook-associated proteins were induced on the filter surface. The gene encoding the anti-sigma-28 factor FlgM was also induced more than 2-fold. Genes encoding the flagellar motor proteins MotA, MotB, MotC, and MotD were induced as well. While it would be expected that planktonic cells of *P. syringae* would be motile, higher levels of expression of motility genes on leaves compared to that in broth cultures has been previously noted [26] and would likely require higher levels of flagellar production and repair [9].

**Lipopolysaccharide synthesis and transport.** All the genes with significant differential expression that are involved in lipopolysaccharide (LPS) synthesis and transport were induced in cells immobilized on filters except *arnB* and *arnA*. Most of these induced genes are involved in LPS transport and lipid A biosynthesis.

**Energy generation.** Many genes encoding proteins involved in oxidative phosphorylation were expressed at a higher level on filter surfaces than in broth cultures. Such genes included

*cyoA*, *cyoB*, *cyoC*, and *cyoD*, all of which encode cytochrome c oxidase subunits, as well as *ccoN*, *ccoO*, and *ccoP* which encode cytochrome c oxidase cbb3-type subunits. Many genes encoding F0F1 ATP synthase subunits were induced as well.

**Transcription.** Similar to that seen for nucleotide synthesis, nearly all of the genes involved in transcription that were differentially expressed were induced on filters compared to that in broth cultures, with the exception of Psyr_4263. Genes up-regulated on surfaces included those encoding the transcription termination factor Rho, the transcription elongation factors GreB and GreA, the transcription anti-termination proteins NusB and NusG, and the DNA-directed RNA polymerase subunits RpoA, RpoC, and RpoB.

**Chemosensing and chemotaxis.** Many genes involved in chemosensing and chemotaxis were induced in cells attached to on the filter surface. Many of the induced genes encoded histidine kinases. Interestingly, Psyr_1306, Psyr_1307 and Psyr_1308, which encode homologs to WspD, WspE, and WspF respectively, were all down-regulated on the filter surface. Psyr_1309, which encodes the homolog for WspR, was also down-regulated. However, this latter gene was assigned to the cyclic diguanylate (cyclic di-GMP) cyclase proteins functional group.

**Replication and DNA repair.** Most of the genes involved in replication and DNA repair were induced on the filter surface. This included Pysr_1408, Psyr_1409, and Psyr_1410 which encode RuvC, RuvA, and RuvB respectively. These genes are also involved in homologous recombination in addition to DNA repair.

**Iron-sulfur proteins.** Many genes encoding iron-sulfur proteins were induced on the filter surface. This included *dsbE* which is involved in cytochrome synthesis.

**Peptiodoglycan/cell wall polymers.** All of the genes involved in encoding peptidoglycan/cell wall polymers that were differentially expressed on filters compared to broth cultures were induced on the filter surface. Most of these genes are involved in peptidoglycan biosynthesis.

**Terpenoid backbone synthesis.** Many genes involved in terpenoid backbone synthesis were induced in cells applied to filter surfaces. This included genes that are part of the deoxyxylulose pathway of terpenoid biosynthesis.

**Quaternary ammonium compound metabolism and transport.** Surprisingly, all genes with differential expression that are required for quaternary ammonium compound (QAC) metabolism and transport were repressed on the filter surface. This included genes encoding proteins involved in glycine betaine, choline, and carnitine metabolism and transport. The gene *betI*, encoding the transcriptional regulator of choline degradation, was also down-regulated. Similar to that seen for genes involved in QAC metabolism, all of the genes that had significant differential expression that are involved in compatible solute synthesis were repressed on the filter surface. Many of these genes contribute to either trehalose or N-acetylglutaminyl-glutamine amide (NAGGN) synthesis. These compounds are solutes that are part of the cellular response to water stress.

**Carbohydrate metabolism and transport.** Many of the genes involved in carbohydrate metabolism and transport were repressed in immobilized cells on filters. This included genes involved in trehalose, mannose, fructose, ribose, arabinose, maltose, manitol, and sorbitol transport as well as genes involved in the pentose phosphate pathway.

**Organic acid metabolism and transport.** Many of the genes involved in organic acid metabolism and transport were also repressed on the filter surface. This included the genes *phnF*, *phnG*, *phnH*, *phnI*, *phnJ*, *phnK*, *phnL*, *phnM*, *phnN*, and *phnP* which are all involved in phosphonate metabolism and transport. The transcriptional regulator of vanillate metabolism, *vanR*, was also down-regulated.

**Phytotoxin synthesis and transport.** All of the differentially expressed genes involved in phytotoxin synthesis and transport were repressed in cells applied to filters. These genes

included *salA*, which is the regulator of syringomycin, as well as *slyA*, the regulator of syringolin A production. Other genes involved in syringolin synthesis and transport (*slyB*, *slyC*, *slyD*, and *slyE*), syringomycin synthesis and transport (*syrE*, *syrC*, *syrB1*, *syrP*, and *syrD*), and syringopeptin synthesis and transport (*sypA*, *sypB*, and *sypC*). Syringomycin and syringopeptin secretion proteins PseA and PseB were also down-regulated in immobilized cells on filters.

**Amino acid metabolism and transport.**   Many of the genes involved in amino acid metabolism and transport were repressed on the filter surface. This included genes involved in gamma-aminobutyric acid (GABA) metabolism (*gabT-2*, *gabD-2*, *gabT-1*, *gabD-1*, *gabD-3*, and *gabP*).

**Secondary metabolism.**   All of the differentially expressed genes involved in secondary metabolism were repressed on the filter with the noteworthy exception of Psyr_2575, Psyr_2576, and Psyr_2577 which encode SyfR, SyfA, and SyfB, responsible for the regulation of and production of syringafactin, respectively. It was therefore intriguing to find that syringafactin production was the sole example of secondary compounds that were not down-regulated when cells transitioned from a planktonic to a sessile state.

## Discussion

The remarkably strong and rapid induction of *syfA* in *P. syringae* B728a in cells transferred from broth culture to any of several different types of surfaces encouraged us to test the hypothesis that a variety of other traits would exhibit similar surface-dependent changes in expression. The rapidity with which *syfA* induction occurred on all of the various surfaces makes it unlikely that cells modified their local microhabitat in any substantial way. We thus presume that assessment of transcriptional changes within two hours after transfer to these surfaces reflected contact-specific gene regulation rather than changes responsive to altered microenvironments that have been seen in studies of biofilms. Such studies typically examined bacteria many hours after attachment, usually after a thick biofilm had formed on surfaces [27–29]. We also presume that regulatory shifts that occurred shortly after contact with a surface also were distinct from those conditioned by other secondary events such as cell-cell contact or cell density-dependent regulatory processes. Given the brief period of immobilization of cells on a filter surface, we were surprised to find such a high proportion of the genes in *P. syringae* B728a to respond to this transition. In addition to induction of the genes conferring syringafactin production and its regulation (*syfA*, *syfB*, and *syfR*), it was striking that genes involved in many other cellular processes such as flagellar synthesis and motility, LPS synthesis and transport, chemosensing and chemotaxis, siderophore synthesis and transport, and DNA replication and repair underwent changes in expression. While coherent arguments could be made for why some of these processes should exhibit contact-dependent expression, the responses of many other processes remain enigmatic. The induction of many genes required for the synthesis of peptidoglycan/cell wall polymers and LPS synthesis and transport seen in the immobilized cells (Table 1) suggests that, like that of other bacteria for which surface contact-dependent gene expression has been investigated [30, 31], *P. syringae* may also utilize a mechanosensitive pathway for sensing a surface that involves disruption and repair of the cell envelope. Previous studies have suggested that cells experience cell wall damage when they contact a surface [8, 30]. Peptidoglycan and cell wall synthesis might be induced to repair any damage that resulted from such an encounter. Moreover, the bacterial outer membrane also contains abundant lipopolysaccharides that could also be disrupted during physical binding of cells to a hard surface, making LPS synthesis essential [9, 32, 33]. Interestingly, this initial cell wall stress may also be a cue for the differential expression of other surface-regulated genes as suggested by others [8].

Historically, it has been common to study bacteria in broth cultures. While it is presumed that such cultures are more homogeneous and facilitate studies of coordinated patterns of gene expression etc., many aspects of the manner in which microorganisms interact with their environment cannot be studied in such a setting [8, 11, 12]. For example, low nutrient conditions in broth cultures are typically associated with both high cell densities and low oxygen levels. In contrast, cells on the leaf surface would typically experience low nutrient conditions in a fully aerobic environment. Although *P. syringae* B728a can be found in aquatic environments, it also colonizes the surface of leaves, a decidedly different environment where it encounters both low nutrient concentrations and high oxygen levels [1, 3, 34]. Furthermore, the study of bacterial adaptations to life on surfaces has focused almost entirely on the process of biofilm formation in aquatic settings [27–29]. The three-dimensional aggregates of bacteria that often form in such aqueous environments, in which nutrients are provided by flowing liquids, are almost certainly very different from the monolayers of bacteria that typically develop on leaf surfaces [35, 36]. Delivery of soluble nutrients in flowing liquids is probably an exceptional situation on leaves [2, 3]. In addition, biofilm formation in a flowing liquid environment occurs over relatively long periods of time. Cells are acquired by, or develop within, a biofilm over a long period of time, during which the nature of the environment within the biofilm changes dramatically, with large spatiotemporal variations [27]. For both practical and other reasons, studies of aquatic biofilms typically have examined gene regulation only 24 hours or later after biofilm initiation [27, 29]. Few studies have addressed very early stages of biofilm formation such as this study.

Genes involved in both transcription and translation were typically induced upon cell immobilization (Table 1). Given that the transcription of a large number of genes was rapidly increased upon transitioning from a planktonic to a sessile state (Figs 3 and 4), there would need to be a corresponding increase in translation to produce the corresponding proteins. Likewise, genes linked to nucleotide metabolism and transport, transcription, and replication and DNA repair (Table 1) were also induced upon immobilization of cells. It would follow that higher rates of nucleotide synthesis or procurement would be needed to support RNA synthesis associated with the elevated transcription. The apparent elevated DNA replication associated with cell immobilization would also require increased expression of genes enabling nucleotide metabolism. Studies have suggested that cells on leaf surfaces and attached to apoplastic surfaces experience oxidative stresses resulting from plant defenses [37–39]. These compounds, including hydroxyl radicals created when iron reacts with $H_2O_2$, can be damaging to iron-sulfur proteins and DNA [39]. Genes involved in iron-sulfur protein synthesis and DNA repair would be expected to be induced on surfaces, as we observed here, if cells anticipated such a chemical assault on plant surfaces and linked their expression with surfaces per se.

Genes required for siderophore synthesis and transport as well as iron metabolism and transport, typically activated under conditions of low iron availability [26] were also induced in immobilized cells (Table 1), suggesting that iron was less available to cells on the filter surface than in broth media. Access to the limited iron in any environment would be expected to be determined by its delivery to the vicinity of the cell by diffusion. Such a process would be diminished in the two-dimensional environment on the surface of filters compared to the three-dimensional habitat of cells immersed in a liquid medium. Cells affixed to a surface also would be unable to move in response to local resource depletion to gain access to such molecules [13]. Given that iron is commonly present in low amounts on the natural surfaces that *P. syringae* typically inhabits, such as leaves [40, 41] immobilization of cells on filters might have mimicked this physical cue to anticipate transition into such low-iron environments.

The many genes involved in flagellar synthesis and motility and chemosensing and chemotaxis that were surface induced (Table 1) would be important in the colonization of leaf surfaces that harbor limited amounts of nutrient resources that are also heterogeneously dispersed [2, 42, 43]. Moreover, pathogenic bacteria such as *P. syringae* often eventually colonize the leaf apoplast—a process requiring cells to move towards and enter a stomata or breaks in the cuticular surface of leaves to access this intercellular habitat [34]. The up-regulation of motility and chemotaxis genes was also observed in a study examining the transcriptome of *P. syringae* B728a on leaf surfaces and apoplast when compared to that in a liquid medium *in vitro* [26]. Direct support for the importance of cell motility of *P. syringae* on leaves was provided by studies of Haefele and Lindow [43] who showed that non-motile mutants were much less fit than motile cells.

Surprisingly, genes involved in QAC metabolism and compatible solute synthesis tended to be repressed in cells on filter surfaces (Table 2). Such compounds often serve as compatible solutes whose purpose is to maintain equilibrium of water availability between the inside of the cell and the outside environment. Thus, genes involved in compatible solute synthesis would be expected to be up-regulated in cells experiencing osmotic or matric stresses [26]. Previous studies have suggested that the leaf surface is often sufficiently dry that compatible solute synthesis is needed to combat matric stress [44, 45]. However, since this experiment was performed on filters placed on nutrient agar, the water status of cells upon such a surface is uncertain. While one might imagine such a surface to be a drier habitat than what cells suspended in the corresponding broth medium might experience, it is likely that this surface would be moister than leaf surfaces typically are. The disruption of the gel matrix that would occur by application of a filter on an agar surface might release free water in a process known as syneresis [46]. Indeed, filters immediately appeared transiently wet after application to the agar surface. While the leaf surface is composed of a waxy, hydrophobic cuticle to prevent the release of water vapor [47, 48], a membrane filter is by definition quite porous and would enable the movement of water onto the filter surface where it could wet cells.

The down-regulation of genes encoding carbohydrate metabolism and transport, organic acid metabolism and transport, and amino acid metabolism and transport (Table 2) upon immobilization of cells was surprising. While we expected nutrients to move from the agar matrix through the filter to the attached cells on top, it is likely that the rate at which nutrients are replenished by this diffusional process would be slower than that occurring during mixing of cells in a planktonic state in the corresponding shaken broth medium. It might thus be expected that the immobilized cells could experience a locally more nutrient-limited environment than those of the planktonic cells. The common observation of slower growth of bacteria on the surface of membranes placed on agar media than in shaken broth of the same composition also supports such a model. Interestingly it has been noted that many genes in *P. syringae* involved in amino acid metabolism and transport were repressed on the leaf surface [26].

Surprisingly, genes involved in phytotoxin synthesis and transport as well as many other genes enabling production of other secondary metabolites were repressed on the filter surface (Table 2). Such compounds often act as virulence factors. We had expected that contact with a surface would serve as a cue to *P. syringae* that it had encountered a potential host plant after transitioning from a planktonic existence, and thus one would have expected such compounds to be expressed. However, the large majority of *P. syringae* cells in the phyllosphere of healthy leaves usually occur only as epiphytes on the surface of leaves rather than within the apoplast [49]. Thus, when *P. syringae* colonizes the leaf surface it seldom interacts directly with living plant cells and thus would not be expected to benefit from the production of various toxins. Rather, such toxins would prove beneficial only after it had entered the leaf apoplast [34] and their expression is highly elevated in the apoplast, but not on the leaf surface [26].

Given that surface sensing by bacteria is thought to involve interactions of cell appendages such as pili with surfaces followed by reinforcement of any binding by the production of extracellular polymers [10, 11, 50], we had expected that pili and EPS genes might well have been responsive to surface attachment. However, when we analyzed the genes belonging to the functional category of "pili synthesis and regulation," less than half of the genes were differentially expressed. We assessed the hypergeometric distribution of these genes' expression to see if there was a significant portion of these genes that were either up-regulated or down-regulated. However, the p-values that we calculated did not support a significant effect. For EPS genes, we looked at the expression of these genes individually, but we did not see any significant trends. For instance, the p-values did not provide any support for the differential regulation of genes involved in the biosynthesis of the Psl polysaccharide upon immobilization. The p-values did tend to be significant for the genes involved in the biosynthesis of alginate, but most of these genes were not significantly up-regulated nor down-regulated. This is most likely because the time period in which the cells were attached to a surface was too short for the genes involved in exopolysaccharide synthesis that might strengthen surface attachment to be induced.

Since gene expression of **P. syringae** B728a had previously been compared between broth cultures and cells recovered after growth on leaves for 3 days [26], it was interesting to compare patterns of differential gene expression with that seen in this study. It was noteworthy that we did not find a similar pattern of gene expression between the two studies (S2 Fig). We reasoned that this is because the bacterial cells in the earlier report had grown on leaf surfaces for 3 days before gene expression was assessed and that differential gene expression was assessed by comparing against the inoculum of the cells that had been grown in a minimal broth medium. Such cells thus had a very long period of time to respond to, and probably change, the leaf microenvironment. The gene expression in these cells would most likely be more similar to cells found in a biofilm as opposed to cells attached to a surface for only 2 hours and also needed to respond to the diverse nutrients present on leaves. The chemical environment on the leaf was certainly much more diverse than the minimal medium for which comparison was drawn, and we would expect that the bacteria strongly modulated gene expression to access these resources. Given this, we also made direct comparisons of our results with those few other studies that have also looked at the more immediate changes in gene expression that occur upon attachment to a surface. One of the few studies that have examined gene expression in bacterial cells soon after surface attachment assessed gene expression in *Escherichia coli* strain CSH50 by microarray analysis one, four, and eight hours after planktonic cells had attached to mannose agarose beads submerged in cultures [13]. This study revealed that changes in gene expression upon immobilization of cells were both very rapid and often transient as many genes that had become induced by one hour of surface attachment exhibited subsequent decreases in expression by four hours of attachment. By eight hours of attachment many of these genes exhibited continued decreases in expression while others only then became induced. Interestingly, we found similar functional genes in *P. syringae* to be up-regulated soon after cells attached to a surface, including *emrB* encoding a drug resistance transporter as well as other genes in this operon, and *glnH*, involved in the transport of glutamine (S3 Fig). The gene *marA*, involved in resistance to a variety of antimicrobial compounds and *marR* encoding its regulator as well as *ahpF*, *grxA*, and *katG* and their regulator *oxyR*, collectively mediating oxidative stress tolerance were also up-regulated in both systems (S3 Fig). Genes involved in attachment and DNA repair were up-regulated in response to attachment in both systems as well. Taken together, these results suggest that both *E. coli* and *P. syringae* experience chemical stresses to which these defensive responses are necessary soon after mobilization on surfaces. Alternatively, surface attachment is taken as a cue to alert

cells that they have arrived at a surface upon which such chemical stresses could be anticipated. *E. coli* can experience chemical stresses associated with innate resistance responses of animal cells [51] as well as plants [37, 39]. Likewise, *P. syringae* experiences defensive chemicals as well as oxidative stress in both compatible and incompatible host plants [38, 52] and induces defensive genes in response [26]. Our work suggests that many such traits might be induced in *P. syringae* merely upon contact with the leaf surface itself. Thus, stress response traits, presumably less important in an open, aquatic environment, are rapidly induced in both *P. syringae* and *E. coli* when host defenses might otherwise prove lethal. An alternative model in which both organisms are simply anticipating a biofilm lifestyle is also attractive since such a behavior would be less habitat-specific. In this way, these dissimilar bacteria would have similar responses to the changing nature of a biofilm habitat that could occur in a variety of biotic and abiotic settings.

In another study, Siryaporn *et al.* [31] examined rapid differential expression of genes in *Pseudomonas aeruginosa* UCBPP-PA14 one hour after attaching to a surface, finding that *pilY1* plays a central role in its surface-dependent induction of virulence and that the expression of many genes was rapidly changed. Despite finding little surface-mediated induction of this homolog in *P. syringae*, the homologs of several other genes were found to be induced in *P. syringae* (S3–S6 Figs). Among these genes were those involved in flagellar synthesis and motility. A variety of genes involved in cell wall synthesis also exhibited similar regulatory responses to surfaces in both bacteria. The genes *mraY* and *murA*, involved in peptidoglycan synthesis, as well as *lpxC* and *lpxD*, involved in LPS synthesis, were induced upon immobilization of both *P. syringae* and *P. aeruginosa* (S6 Fig), suggesting that cell wall damage is common upon interacting with surfaces [8, 30, 53]. Likewise, *ispZ*, involved in cell division, was also induced on surfaces in both studies (S6 Fig). It thus appears that there are many common features of the initial interaction of a variety of bacteria taxa with surfaces, and that the nature of the surface with which they interact may play a subservient role in dictating such responses. A better understanding of the initial interaction of bacteria with surfaces hopefully will lead to better manage their interactions in both agricultural and biomedical settings [12].

## Supporting information

**S1 Table. Number of sequencing reads.** Number of sequencing reads obtained for the various replicate samples of *Pseudomonas syringae* B728a cells recovered 2 hours after application to filters or from cells in broth cultures used to inoculate the filters.
(PDF)

**S1 Fig. Smear plot documenting the differential expression of genes in *Pseudomonas syringae* B728a.** Smear plot documenting the differential expression of genes in *Pseudomonas syringae* B728a 2 hours after inoculation onto filter surfaces compared to that in planktonic cells as a function of their levels of expression. Shown is the proportion of the genes that were differentially expressed on the filter surface (red) and those not differentially expressed on a surface (black).
(PDF)

**S2 Fig. Relationship between magnitude (fold change) of differential expression of genes in *Pseudomonas syringae* B728a on a filter and in *P. syringae* B728a on a leaf surface.** Differential gene expression of genes of *P. syringae* B728a in cells recovered after growth on bean leaf surfaces for 3 days compared with that in a minimal broth medium in the study of Yu *et al.* [26] with that of the differential expression of those genes in a rich broth medium compared with that of cells transferred to a membrane for 2 hours in this study. Shown is the $\log_2$

fold differential expression in each study. Note that the expression of 5 genes having exceptionally large differential expression in at least one setting has been omitted to enable better illustration of the remaining genes.
(PDF)

**S3 Fig. Comparison of differential expression of homologs of genes in *Pseudomonas syringae* B728a and in *E. coli* CSH50.** Comparison of differential expression of homologs of genes in *Pseudomonas syringae* B728a 2 hours after application to membrane surfaces and of *E. coli* CSH50 1 hour after attachment to mannose agarose beads in the study of Bhomkar *et al*. [13]. Differential up-regulation of genes is shown in green, down-regulation is depicted in red, and no change in expression is shown in yellow.
(PDF)

**S4 Fig. Relationship between magnitude (fold change) of differential expression of gene homologs in *Pseudomonas syringae* B728a and in *E. coli* CSH50.** Relationship between magnitude (fold change) of differential expression of gene homologs in *Pseudomonas syringae* B728a 2 hours after transfer to a filter surface and that of *E. coli* CSH50 1 hour after attachment on agarose beads in the study of Bhomkar *et al*. [13].
(PDF)

**S5 Fig. Comparison of the number of gene homologs in *Pseudomonas syringae* B728a and in *Pseudomonas aeruginosa* UCBPP-PA14.** Comparison of the number of gene homologs in *Pseudomonas syringae* B728a (blue) and *Pseudomonas aeruginosa* UCBPP-PA14 (red) that were up-regulated when transferred to filter surfaces for 2 hours or were attached to a glass surface for 1 hour in the study of Siryaporn *et al*. [31], respectively. Using the hypergeometric distribution, this overlap was determined to be significant with a p-value of 1.65E-05 [22].
(PDF)

**S6 Fig. Comparison of differential expression of homologs of genes in *Pseudomonas syringae* B728a and in *Pseudomonas aeruginosa* UCBPP-PA14.** Comparison of differential expression of homologs of genes in *Pseudomonas syringae* B728a 2 hours after application to membrane surfaces and of *Pseudomonas aeruginosa* UCBPP-PA14 wild-type cells attached to a glass surface for 1 hour in the study of Siryaporn *et al*. [31]. Differential up-regulation of genes is shown in green, down-regulation is depicted in red, and no change in expression is shown in yellow.
(PDF)

## Acknowledgments

We thank Kevin Hockett and Gwyn Beattie for providing the gene functional category assignments for *Pseudomonas syringae* B728a. We also thank Rebecca Mackelprang and Ke Bi for their assistance with R to analyze RNA sequencing results. We also thank Steven Ruzin and Denise Schichnes for assistance in Biological Imaging. We really appreciate the helpful comments made by 2 referees in review.

## Author Contributions

**Conceptualization:** Monica N. Hernandez, Steven E. Lindow.

**Data curation:** Monica N. Hernandez.

**Formal analysis:** Monica N. Hernandez.

**Funding acquisition:** Monica N. Hernandez, Steven E. Lindow.

**Investigation:** Monica N. Hernandez.

**Methodology:** Monica N. Hernandez.

**Project administration:** Steven E. Lindow.

**Resources:** Steven E. Lindow.

**Supervision:** Steven E. Lindow.

**Validation:** Monica N. Hernandez.

**Visualization:** Monica N. Hernandez.

**Writing – original draft:** Monica N. Hernandez.

**Writing – review & editing:** Monica N. Hernandez, Steven E. Lindow.

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
