## [Decision Letter · Decision Letter 0]

26 Nov 2020

PONE-D-20-32681

Contact-dependent traits in *Pseudomonas syringae* B728a

PLOS ONE

Dear Dr. Hernandez,

Thank you for submitting your manuscript to PLOS ONE. After careful consideration, we feel that it has merit but does not fully meet PLOS ONE’s publication criteria as it currently stands. Therefore, we invite you to submit a revised version of the manuscript that addresses the points raised during the review process.

As you can see in the comments by both reviewers there are several issues that need adjustments. For instance there is a lack of statistical analyses. Except for the verification of the RNA Seq data by qRT-PCR (optional) I strongly suggest to modify the manuscript according to the comments of both reviewers.

We look forward to receiving your revised manuscript.

Kind regards,

Günther Koraimann

Academic Editor

PLOS ONE

Journal Requirements:

Reviewers' comments:

Reviewer's Responses to Questions

**Comments to the Author**

1. Is the manuscript technically sound, and do the data support the conclusions?

Reviewer #1: Yes

Reviewer #2: Yes

2. Has the statistical analysis been performed appropriately and rigorously? 

Reviewer #1: No

Reviewer #2: I Don't Know

3. Have the authors made all data underlying the findings in their manuscript fully available?

Reviewer #1: Yes

Reviewer #2: Yes

4. Is the manuscript presented in an intelligible fashion and written in standard English?

Reviewer #1: Yes

Reviewer #2: Yes

5. Review Comments to the Author

Reviewer #1: The present work showed that expression of syfA is induced in Pseudomonas syringae B728a cells immobilized on surfaces. RNA-seq analysis revealed that more 3000 deferentially expressed genes between planktonic cells and immobilized cells. The techniques used in this study were sound, but additional verification and analyses are suggested to improve this manuscript.

1. P values are missing all figures.

2. Key genes/categories should be labelled and highlighted in Figure 3 and 4.

3. Part of the RNA-seq data should be verified by qRT-PCR.

4. Short paragraphs should be combined.

5. Suggest to add a final figure to summarize all changed gene categories, which would help readers to better understand the overall contact-dependent traits in P. syringae B728a.

Reviewer #2: This manuscript explores the question of how Pseudomonas syringae bacteria respond to surface contact. The work follows up on an observation that the expression of a gene for a key biosurfactant in this organism noticeably responded to surface contact, leading the authors to wonder if other genes responded similarly. Given the lifestyle of this organism as a resident on leaf surfaces, they hypothesized that the bacteria use surface contact as one mechanism by which they initiate the expression of physiological traits needed to live on leaves. The manuscript provides data demonstrating that this biosurfactant gene is induced within two hours following surface contact, that this induction occurs on a variety of distinct types of surfaces including leaves, and that large sets of genes are upregulated and downregulated following surface contact when examined after two hours. Much of the manuscript speculates on biological explanations for the upregulation and downregulation of genes. The work is sound and clearly presented, and the explanations are credible.

The major interesting finding is the sheer number of genes that were influenced by the surface contact treatment, with a few surprises in the nature of the genes that were altered. The data confirmed the increased expression of the biosurfactant genes on a surface, but surprisingly indicated decreased expression of all other secondary metabolite genes as well as a range of genes involved with osmotic stress tolerance. Similarly surprising, the data indicated increased expression of flagellar synthesis genes and siderophore genes. These unexpected findings make the results quite interesting to contemplate.

A major consideration is the extent to which the results could reflect unintended consequences of the treatment. This was one area that I thought warranted a little more discussion. The gene expression comparisons were made between cells kept in liquid culture, and cells from that culture that were put on a membrane, subjected to vacuum-mediated removal of liquid for 5 seconds, and then transferred onto the surface of a KB plate for 2 hours. Thus, as compared to the cells maintained in broth, the treated cells experienced a 2-hour period of immobilization on a membrane, as well as exposure to agar in the KB plate and a 5-second period of potential shear stress and oxygen exposure during the vacuum treatment. Was the vacuum application enough to induce a mechanical stress? The authors explain how contact-mediated cell envelope/cell wall stress may explain the induction of genes for LPS and peptidoglycan synthesis. Could the mechanical stress due to the vacuum have been enough to alter the mechanosensitive ion channels? This has been predicted (Kimkes and Heinemann 2020 How bacteria recognise and respond to surface contact FEMS Microbiol Rev), and I wondered if this may have resulted in sufficient inward water movement to repress the expression of genes involved in osmotic stress tolerance. Similarly, could this have been sufficient to remove flagella, thus inducing flagellar synthesis because of loss? (This assumes that the cells in liquid media had flagella to begin with, which may or may not have been the case.)

The description of how they identified genes as differentially regulated (lines 159-161) does not mention that an approach was taken to account for the large number of comparisons that were made (generally a false discovery rate is stated). Such adjustments are mentioned in the statistical analysis section for the hypergeometric distribution work, but it is not clear if they were accounted for in identifying differentially regulated genes.

The authors speculate that this transition to surface growth may be anticipatory of what is involved in growing a leaf surface. Since co-author Lindow previously looked at the induction of genes in this same strain on leaves (ref 26), then it would have been useful to have a more explicit comparison of the genes expressed in this study to those induced on leaves. For example, even a Venn diagram like that shown for P. aeruginosa in Fig S6 would have been useful. (Note that this comparison was done for some selected genes throughout the discussion.)

The text describing Fig 1 (lines 175-178) does not align well with the legend within the figure. Is the “filter on plastic” the third treatment on line 177 (“a 0.4 um Isopore filter floated on the surface of a small quantity of KB broth”)? The text in the legend (Lines 193-194) needs a clear description of what these treatments are, particularly as the text only mentions “these surfaces” but broth is not a surface.

Figure 2 would be improved with the inclusion of statistics in the figure or legend. It is not clear that statistics were performed to support the conclusion that the induction was not influenced by the nature of the surface.

Fig S1 would be useful as a table. Fig S2 is not useful. I also didn’t find Figs S4-S7 particularly useful.

Fig 3 needs to have a description of what the dendograms are on the x and y axes.

Just a few comments on the findings presented:

The literature describing surface sensing by bacteria generally involves the idea that as planktonic cells move toward a surface, they experience changes in the physical and chemical properties of the surface, attach via cell appendages, and then attach more firmly via other factors such as extracellular polymers. The treatment used here did not allow for movement, but could have allowed for these stages to occur. The authors do not mention pili genes, which could be involved in attachment, or EPS genes (e.g., the genes for Pel, which increases adhesion of P. aeruginosa to surfaces, and are present in this P. syringae strain). Were these genes affected?

The authors propose that “a secondary signal, indicative of conditions within the leaf, and not a surface itself, is required for expression of such virulence factors” as phytotoxins. Whereas this is possible, this does not explain why their expression would be lower on a filter on KB than in KB broth, since broth would also lack such secondary signals. The authors may want to add a qualification to their statement.

The discussion in the paragraph comparing the results with P. syringae to those with E. coli (lines 526-550) suggest that both bacterial genera may be expressing genes in response to a surface in anticipation of needing particular traits for survival on a host. Although appealing, I find it less credible than they are anticipating traits for survival in a biofilm. Both organisms live in a diversity of habitats, and inducing genes particular to survival on a host when contacting debris in sewage (E. coli) or in streams (P. syringae) could be wasteful; however, growth in a biofilm would occur more universally on the surfaces they encounter, host or otherwise.

Minor editorial comments

Line 62 Consider replacing “expanding the conditions under which” with “expanding the time frame during which”

Lines 130-143 If the RNA isolation followed the RNA kit instructions, then this could be shorted to simply state this (or state this and describe modifications).

Lines 268 and 282 These should state “up-regulated/down-regulated in immobilized cells as compared to planktonic cells…”.

Line 442 Spaciotemporal should be spatiotemporal

6. PLOS authors have the option to publish the peer review history of their article (what does this mean?). If published, this will include your full peer review and any attached files.

Reviewer #1: No

Reviewer #2: No

---

## [Author Response · Author response to Decision Letter 0]

10 Jan 2021

Editors comments:

Thank you for submitting your manuscript to PLOS ONE. After careful consideration, we feel that it has merit but does not fully meet PLOS ONE’s publication criteria as it currently stands. Therefore, we invite you to submit a revised version of the manuscript that addresses the points raised during the review process.

As you can see in the comments by both reviewers there are several issues that need adjustments. For instance there is a lack of statistical analyses. Except for the verification of the RNA Seq data by qRT-PCR (optional) I strongly suggest to modify the manuscript according to the comments of both reviewers.

We appreciate the generally very positive comments on the submission from both you and the 2 referees. We have made extensive changes to the manuscript that address the comments made by the reviewers. We have added additional statistical analyses and have clarified some points that led to come concerns raised by the referees. We have responded to each comment below.

Reviewer #1 (Comments for the Author)

The present work showed that expression of syfA is induced in Pseudomonas syringae B728a cells immobilized on surfaces. RNA-seq analysis revealed that more 3000 deferentially expressed genes between planktonic cells and immobilized cells. The techniques used in this study were sound, but additional verification and analyses are suggested to improve this manuscript.

1. P values are missing all figures.

We have performed Tukey-adjusted post-hoc testing for Figure 2 to indicate which groups are different. Significance was determined by a p-value of less than 0.05.

2. Key genes/categories should be labelled and highlighted in Figure 3 and 4.

We agree and have added gene labels to Figures 3 and 4.

3. Part of the RNA-seq data should be verified by qRT-PCR.

While this might have been useful, the state of the art of RNA sequencing is currently such that such verification is not essential. We unfortunately are also not in a position to perform qRT-PCR at this time.

4. Short paragraphs should be combined.

This is a helpful suggestion and have made these changes throughout to merge those paragraphs that address a similar issue.

5. Suggest to add a final figure to summarize all changed gene categories, which would help readers to better understand the overall contact-dependent traits in P. syringae B728a.

This is a very useful suggestion, as we now realize that there was no one figure that would allow readers to gain insight into the relative contributions of the various gene categories to the extensive differential gene expression we observed. We have now prepared a figure (now included as Figure 5) that shows, for those gene categories for which there is statistical support for differential gene expression, the proportion of the genes in that gene category that were either significantly up-regulated (Figure 5A) or down regulated (Figure 5B).

Reviewer #2 (Comments for the Author):

This manuscript explores the question of how Pseudomonas syringae bacteria respond to surface contact. The work follows up on an observation that the expression of a gene for a key biosurfactant in this organism noticeably responded to surface contact, leading the authors to wonder if other genes responded similarly. Given the lifestyle of this organism as a resident on leaf surfaces, they hypothesized that the bacteria use surface contact as one mechanism by which they initiate the expression of physiological traits needed to live on leaves. The manuscript provides data demonstrating that this biosurfactant gene is induced within two hours following surface contact, that this induction occurs on a variety of distinct types of surfaces including leaves, and that large sets of genes are upregulated and downregulated following surface contact when examined after two hours. Much of the manuscript speculates on biological explanations for the upregulation and downregulation of genes. The work is sound and clearly presented, and the explanations are credible.

The major interesting finding is the sheer number of genes that were influenced by the surface contact treatment, with a few surprises in the nature of the genes that were altered. The data confirmed the increased expression of the biosurfactant genes on a surface, but surprisingly indicated decreased expression of all other secondary metabolite genes as well as a range of genes involved with osmotic stress tolerance. Similarly surprising, the data indicated increased expression of flagellar synthesis genes and siderophore genes. These unexpected findings make the results quite interesting to contemplate.

A major consideration is the extent to which the results could reflect unintended consequences of the treatment. This was one area that I thought warranted a little more discussion. The gene expression comparisons were made between cells kept in liquid culture, and cells from that culture that were put on a membrane, subjected to vacuum-mediated removal of liquid for 5 seconds, and then transferred onto the surface of a KB plate for 2 hours. Thus, as compared to the cells maintained in broth, the treated cells experienced a 2-hour period of immobilization on a membrane, as well as exposure to agar in the KB plate and a 5-second period of potential shear stress and oxygen exposure during the vacuum treatment. Was the vacuum application enough to induce a mechanical stress? The authors explain how contact-mediated cell envelope/cell wall stress may explain the induction of genes for LPS and peptidoglycan synthesis. Could the mechanical stress due to the vacuum have been enough to alter the mechanosensitive ion channels? This has been predicted (Kimkes and Heinemann 2020 How bacteria recognise and respond to surface contact FEMS Microbiol Rev), and I wondered if this may have resulted in sufficient inward water movement to repress the expression of genes involved in osmotic stress tolerance. Similarly, could this have been sufficient to remove flagella, thus inducing flagellar synthesis because of loss? (This assumes that the cells in liquid media had flagella to begin with, which may or may not have been the case.)

This is a good point, and we thank you for pointing out the very new manuscript by Kimkes and Heinemann (2020) which relates to this question. We should clarify that the filters were exposed to the vacuum only long enough for the liquid to be removed. We were cognizant of possible stresses that the cells would have seen if exposed to excessive vacuum and thus were careful to remove the liquid from suspensions applied to the membrane surfaces by application of only a very gentle volume for only the least time needed for liquid removal. While we had noted a 5 second removal process in the original manuscript in reality, this process occurred almost instantly. We have modified the text to reflect this. While it is certainly possible that the water removal process could have resulted in some mechanical stress or to have caused some inward movement of water, we feel this was unlikely or minor at best. This gentle water removal also seemed unlikely to have removed flagella, had they been present. We have now discussed these possible scenarios in the discussion.

The description of how they identified genes as differentially regulated (lines 159-161) does not mention that an approach was taken to account for the large number of comparisons that were made (generally a false discovery rate is stated). Such adjustments are mentioned in the statistical analysis section for the hypergeometric distribution work, but it is not clear if they were accounted for in identifying differentially regulated genes.

We thank the reviewer for pointing out our omission of the description of some of the bioinformatics analyses used. We should have clarified how we accounted for the false discovery rate while identifying differentially regulated genes. We used the Fisher’s exact test to make comparisons between groups. From this test we obtained values for the log fold change and the log counts per million as well as the p-value and the false discovery rate for each gene. We selected only those genes that had a false discovery rate of <0.05. We have now added this explanation to our manuscript.

The authors speculate that this transition to surface growth may be anticipatory of what is involved in growing a leaf surface. Since co-author Lindow previously looked at the induction of genes in this same strain on leaves (ref 26), then it would have been useful to have a more explicit comparison of the genes expressed in this study to those induced on leaves. For example, even a Venn diagram like that shown for P. aeruginosa in Fig S6 would have been useful. (Note that this comparison was done for some selected genes throughout the discussion.)

This is a good point. We had indeed compared gene expression in our cells that had been exposed to a surface for only 2 hours with those that had grown on leaves as described in an earlier report. It was noteworthy that we did not find a similar pattern of gene expression between the two studies. We reasoned that this is because the bacterial cells in the earlier report had grown on leaf surfaces for 3 days before gene expression was assessed and that differential gene expression was assessed by comparing against the inoculum of the cells that had been grown in a minimal broth medium. Such cells thus had a very long period of time to respond to, and probably change, the leaf microenvironment. The gene expression in these cells would most likely be more similar to cells found in a biofilm as opposed to cells attached to a surface for only 2 hours and also needed to respond to the diverse nutrients present on leaves. The chemical environment on the leaf was certainly much more diverse than the minimal medium for which comparison was drawn. Given that these 2 rich datasets exist, we agree that it might be of interest for readers to be able to see the highly divergent patterns of differential gene expression in these two comparison settings. We thus have added a Supplemental Figure (Fig S2) in which we show differential gene expression of genes of P. syringae B728a in cells recovered after growth on bean leaf surfaces for 3 days compared with that in a minimal broth medium in the study of Yu et al. (26) with that of the differential expression of those genes in a rich broth medium compared with that of cells transferred to a membrane for 2 hours in this study. We show the log2 fold differential expression of each gene in each study. We have now added a paragraph in the manuscript that notes the distinct patterns of gene expression in established cells of P. syringae on leaves with those that rapidly change upon contact with a surface and to clarify the choices that we made in terms of comparing studies. Given the substantial differences in these 2 studies of P. syringae, we decided to also make direct comparisons of our results with those few other studies that have also looked at the immediate changes in gene expression that occur upon attachment to a surface. 

The text describing Fig 1 (lines 175-178) does not align well with the legend within the figure. Is the “filter on plastic” the third treatment on line 177 (“a 0.4 um Isopore filter floated on the surface of a small quantity of KB broth”)? The text in the legend (Lines 193-194) needs a clear description of what these treatments are, particularly as the text only mentions “these surfaces” but broth is not a surface.

We agree that our original annotation of the figure was not clear, and have changed the text describing Fig 1 to clarify that the filter was moistened with KB broth and placed onto plastic. We have also changed the notation within the figure and the wording in the legend to Fig 1 clarify the surfaces to which the cells were transferred.

Figure 2 would be improved with the inclusion of statistics in the figure or legend. It is not clear that statistics were performed to support the conclusion that the induction was not influenced by the nature of the surface.

We agree and have performed Tukey-adjusted post-hoc testing in R to better illustrate that induction of syfA was similar on all of the surfaces to which cells were transferred.

Fig S1 would be useful as a table. Fig S2 is not useful. I also didn’t find Figs S4-S7 particularly useful.

We agree that that the supplemental data currently shown in Fig S1 would be more easily viewed as a Table and have made this change. We also agree that the data in Fig S2 is not particularly important and we have removed it. We understand that while not everyone will find Figs S4-S7 particularly useful, we would prefer to retain this data in the supplement since some readers might want to see these visualizations. Also, its inclusion there does not distract from the main presentation in the text.

Fig 3 needs to have a description of what the dendograms are on the x and y axes.

We agree and have added gene names and the treatment conditions to the dendrograms.

Just a few comments on the findings presented:

The literature describing surface sensing by bacteria generally involves the idea that as planktonic cells move toward a surface, they experience changes in the physical and chemical properties of the surface, attach via cell appendages, and then attach more firmly via other factors such as extracellular polymers. The treatment used here did not allow for movement, but could have allowed for these stages to occur. The authors do not mention pili genes, which could be involved in attachment, or EPS genes (e.g., the genes for Pel, which increases adhesion of P. aeruginosa to surfaces, and are present in this P. syringae strain). Were these genes affected?

We appreciate the reviewers understanding of the processes occurring upon surface colonization. We had expected that pili and EPS genes might well have been responsive to surface attachment. However, when we analyzed the genes belonging to the functional category of “pili synthesis and regulation,” less than half of the genes were differentially expressed. We assessed the hypergeometric distribution of these genes expression to see if there was a significant portion of these genes that were either up-regulated or down-regulated, but the p-values that we calculated did not support a significant effect. For EPS genes, we looked at the expression of these genes individually but we did not see any significant trends. For instance, the p-values did not provide any support for the differential regulation of genes involved in the biosynthesis of the Psl polysaccharide upon immobilization. The p-values did tend to be significant for the genes involved in the biosynthesis of alginate, but most of these genes were not significantly up-regulated nor down-regulated. This is most likely because the time period in which the cells were attached to a surface was too short for the genes involved in exopolysaccharide synthesis that might strengthen surface attachment to be induced. We have now included a paragraph in the manuscript to address this.

The authors propose that “a secondary signal, indicative of conditions within the leaf, and not a surface itself, is required for expression of such virulence factors” as phytotoxins. Whereas this is possible, this does not explain why their expression would be lower on a filter on KB than in KB broth, since broth would also lack such secondary signals. The authors may want to add a qualification to their statement.

We agree with the author’s point and that we may have gone too far out on a limb with this argument. We have thus removed this speculation of a secondary signal.

The discussion in the paragraph comparing the results with P. syringae to those with E. coli (lines 526-550) suggest that both bacterial genera may be expressing genes in response to a surface in anticipation of needing particular traits for survival on a host. Although appealing, I find it less credible than they are anticipating traits for survival in a biofilm. Both organisms live in a diversity of habitats, and inducing genes particular to survival on a host when contacting debris in sewage (E. coli) or in streams (P. syringae) could be wasteful; however, growth in a biofilm would occur more universally on the surfaces they encounter, host or otherwise.

We appreciate this comment and agree that a more common anticipation of a biofilm lifestyle that would be less habitat-specific is a likely explanation for similar responses of such dissimilar bacteria. We have now included such a model in our revised manuscript.

Minor editorial comments

Line 62 Consider replacing “expanding the conditions under which” with “expanding the time frame during which”

This is a good suggestion, and we have made this change.

Lines 130-143 If the RNA isolation followed the RNA kit instructions, then this could be shorted to simply state this (or state this and describe modifications).

We agree and have made this change.

Lines 268 and 282 These should state “up-regulated/down-regulated in immobilized cells as compared to planktonic cells…”.

We agree and have made this change.

Line 442 Spaciotemporal should be spatiotemporal

Thanks for catching this. We have made this change.

---

## [Decision Letter · Decision Letter 1]

25 Jan 2021

Contact-dependent traits in *Pseudomonas syringae* B728a

PONE-D-20-32681R1

Dear Dr. Hernandez,

We’re pleased to inform you that your manuscript has been judged scientifically suitable for publication and will be formally accepted for publication once it meets all outstanding technical requirements.

Kind regards,

Günther Koraimann

Academic Editor

PLOS ONE

Additional Editor Comments (optional):

Reviewers' comments:

Reviewer's Responses to Questions

**Comments to the Author**

1. If the authors have adequately addressed your comments raised in a previous round of review and you feel that this manuscript is now acceptable for publication, you may indicate that here to bypass the “Comments to the Author” section, enter your conflict of interest statement in the “Confidential to Editor” section, and submit your "Accept" recommendation.

Reviewer #1: All comments have been addressed

Reviewer #2: All comments have been addressed

2. Is the manuscript technically sound, and do the data support the conclusions?

Reviewer #1: Yes

Reviewer #2: (No Response)

3. Has the statistical analysis been performed appropriately and rigorously? 

Reviewer #1: Yes

Reviewer #2: (No Response)

4. Have the authors made all data underlying the findings in their manuscript fully available?

Reviewer #1: Yes

Reviewer #2: (No Response)

5. Is the manuscript presented in an intelligible fashion and written in standard English?

Reviewer #1: Yes

Reviewer #2: (No Response)

6. Review Comments to the Author

Reviewer #1: (No Response)

Reviewer #2: (No Response)

7. PLOS authors have the option to publish the peer review history of their article (what does this mean?). If published, this will include your full peer review and any attached files.

Reviewer #1: No

Reviewer #2: No

---

## [Editor Report · Acceptance letter]

29 Jan 2021

PONE-D-20-32681R1 

Contact-dependent traits in *Pseudomonas syringae* B728a 

Dear Dr. Hernandez:

I'm pleased to inform you that your manuscript has been deemed suitable for publication in PLOS ONE. Congratulations! Your manuscript is now with our production department. 

Kind regards, 

on behalf of

Dr. Günther Koraimann 

Academic Editor

PLOS ONE